# Quantum storage of entangled photons at telecom wavelengths in a crystal

Ming-Hao Jiang ®[1,4], Wenyi Xue[1,4], Qian He[1], Yu-Yang An[1], Xiaodong Zheng[1], Wen-Jie Xu[1], Yu-Bo Xie[1], Yanqing Lu ®[1], Shining Zhu ®[1] & Xiao-Song Ma ®[1,2,3] ✉

Quantum storage and distribution of entanglement are the key ingredients for realizing a global quantum internet. Compatible with existing fiber networks, telecom-wavelength entangled photons and corresponding quantum memories are of central interest. Recently, $^{167}Er^{3+}$ ions have been identified as a promising candidate for an efficient telecom quantum memory. However, to date, no storage of entangled photons, the crucial step of quantum memory using these promising ions, $^{167}Er^{3+}$, has been reported. Here, we demonstrate the storage and retrieval of the entangled state of two telecom photons generated from an integrated photonic chip. Combining the natural narrow linewidth of the entangled photons and long storage time of $^{167}Er^{3+}$ ions, we achieve storage time of 1.936 µs, more than 387 times longer than in previous works. Successful storage of entanglement in the crystal is certified using entanglement witness measurements. These results pave the way for realizing quantum networks based on solid-state devices.

The quantum internet[1,2]—in synergy with the internet that we use today—promises an enabling platform for next-generation information processing, including exponentially speed-up distributed computation, secure communication, and high-precision metrology. For fiber-based quantum networks, the distance over which entanglement can be distributed is limited to around 100 km, due to losses in the optical fiber. The quantum-communication efficiency can be greatly enhanced, however, by harnessing quantum-repeater architectures[3–7]. The key building block of such architectures is quantum memory, wherein photonic quantum states are stored in quantum states of matter. Quantum memories have been realized with atomic ensembles[8–12], single atoms[13–17], and solid-state systems[18–27]. For the integration of these elements into a quantum network, three general criteria have to be met[7,28,29]: (1) wavelength compatibility with existing telecom networks (i.e., the natural choice for the level transition in the storage device should be around 1.5 µm for low-propagation loss of photons in fiber); (2) long storage times; and (3) multiplexed storage capability. One of the most promising candidates for realizing such a memory are solids doped with rare-earth ions. Several recent experiments on rare-earth ions with long hyperfine lifetimes, such as praseodymium or europium, have established the potential to realize practical memories by demonstrating storage in long-lived spin states[30–32], multimode storage[28,32,33], high efficiencies[34,35] and long storage times[36,37]. However, none of the above-mentioned ions have suitable optical transitions in any of the telecom bands; nonetheless, flying photon at telecom wavelength can be realized either by frequency conversion[10,11,17,38–40] or entanglement heralding with frequency non-degenerate photon-pair sources[19,20,41–45].

The erbium ion has optical transitions in the telecom band around 1536 nm. However, the small ratio of spin lifetime to optical lifetime limits its performance as telecom quantum memories[46,47]. Nonetheless, previous pioneering works showed the storage of entangled states for 5 ns[48], and the storage of heralded single-photon state for about 50 ns[49,50]. Recently, there is a rising interest in isotopically purified rare-earth ions due to their long coherence time compared with unpurified ions[25,51–54]. Rančić et al.[51] showed that in a high magnetic field, $^{167}Er^{3+}$ ions with $I = 7/2$ possess a hyperfine coherence time of 1.3 s and efficient spin pumping of the ensemble into a single hyperfine

[1]National Laboratory of Solid-state Microstructures, School of Physics, College of Engineering and Applied Sciences, Collaborative Innovation Center of Advanced Microstructures, Nanjing University, 210093 Nanjing, China. [2]Synergetic Innovation Center of Quantum Information and Quantum Physics, University of Science and Technology of China, 230026 Hefei, Anhui, China. [3]Hefei National Laboratory, 230088 Hefei, China. [4]These authors contributed equally: Ming-Hao Jiang, Wenyi Xue. ✉e-mail: Xiaosong.Ma@nju.edu.cn

state. This seminal work triggered a recent resurgence of interest in using erbium ions for quantum memory, e.g., for on-chip storage of telecom-band classical light at the single-photon intensity level[55], for efficient initialization using the resolved hyperfine structure for quantum memories[56], and for on-demand storage of weak coherent light with laser-written waveguides[57]. The next milestone of quantum memory in this system is to store quantum entanglement in $^{167}Er^{3+}$ ions and show that entanglement is preserved after storage.

Here, we demonstrate the storage and retrieval of the entangled state of two telecom photons in $^{167}Er^{3+}$ ions. The entangled photon pairs are generated from an integrated photonic chip based on a silicon nitride (SiN) microring resonator, with natural narrow linewidths compatible with $^{167}Er^{3+}$ ions. A storage time of 1.936 μs is achieved, 387 times longer than in previous works. Successful storage of entanglement in the crystal is certified using entanglement witness measurements, with more than 23 standard deviations (−0.234 ± 0.010) at 1.936 μs storage time.

## Results
### Experimental setup
The schematics of our experimental setup is shown in Fig. 1. The storage and retrieval of the entangled state of two telecom photons is realized by implementing following steps: (1) with the entanglement source (Fig. 1a), we create a pair of entangled (signal and idler) photons with central wavelengths in the telecom C-band and narrow bandwidth (~185 MHz) compatible with $^{167}Er^{3+}$ optical transition; (2) we initialize the $^{167}Er^{3+}$ quantum memory (Fig. 1b) and then send the signal photon into the memory for quantum storage up to 1.936 μs; and (3) we analyze the correlation between the retrieved signal photon and the

unstored idler photon with the entanglement analyzer (Fig. 1c), and verify the preservation of entanglement after the quantum storage.

### Entangled photon-pair source
The time-bin entangled photon pairs are generated from an integrated silicon nitride (SiN)[58–65] dual Mach-Zehnder interferometer microring resonator (DMZI-R)[66–69], which is shown in Fig. 2a, using the spontaneous four-wave mixing (SFWM) process (inset of Fig. 2b). As shown in Fig. 1a, the output of a continuous-wave pump laser is chopped by an intensity modulator (IM) into pulses. These pulses are then further amplified with an erbium-doped fiber amplifier (EDFA). Several wavelength-division multiplexers (WDM) are used to clean the spectrum of the pulses, which are subsequently injected into the "In" port of the SiN DMZI-R resonator (Fig. 2a). A pair of telecom-wavelength photons that are non-degenerate in frequency, signal and idler photons, are generated from the resonator and coupled out from the "Drop" port. Signal and idler photons are then separated with a WDM. The remaining pump pulses are coupled out from the "Through" port and detected with a photodiode (PD). We use the detected power signal to monitor the frequency drift between the resonance of the DMZI-R and the pump[65]. The additional "Add" port is used for calibrating the DMZI-R resonance conditions. The unique structure of the DMZI-R source allows us to independently tune the coupling between the waveguides and the ring resonator for the pump pulses, signal, and idler photons. In this way, the pump laser and generated photon pair are coupled at different ports to reduce the on-chip and in-fiber Raman noise generated from the strong pump[66–69]. The SiN DMZI-R resonator chip is both optically and electrically packaged for long-time stable operation.

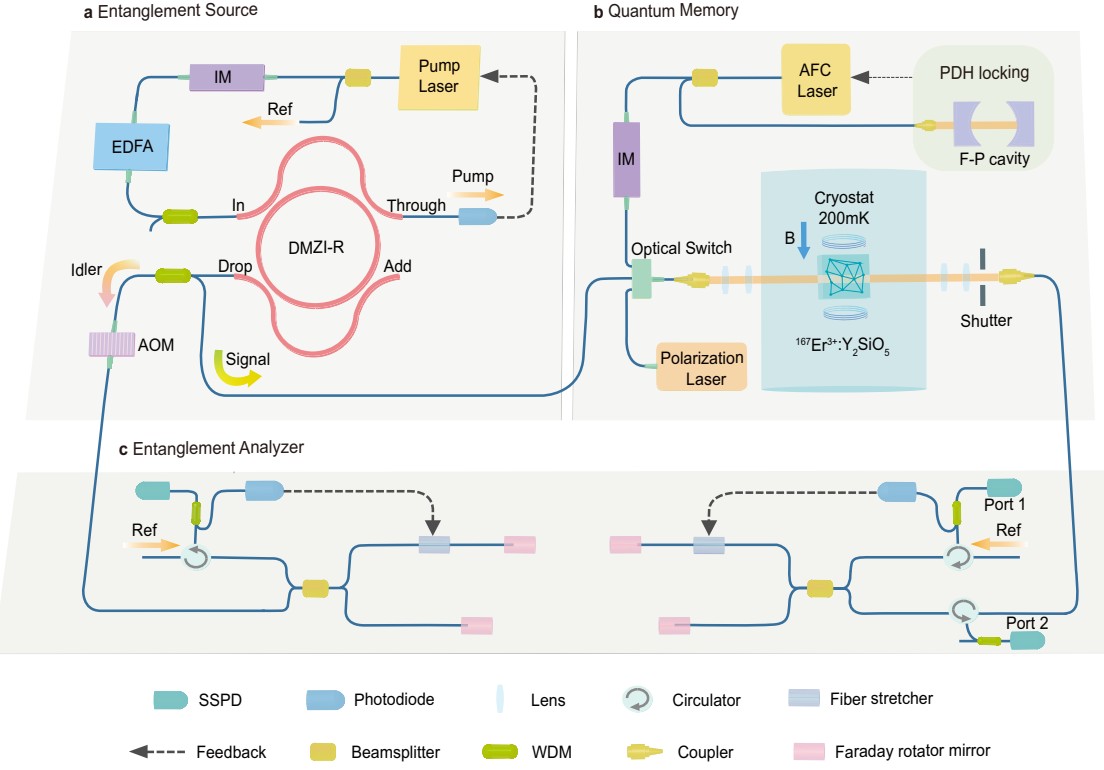

**Fig. 1 | Schematics of the experimental setup. a** Entangled photon pairs are generated with an integrated silicon nitride (SiN) dual Mach-Zehnder interferometer microring resonator (DMZI-R) and separated with a wavelength-division multiplexer (WDM). The microring is pumped by the pump laser followed with an intensity modulator (IM) and an erbium-doped fiber amplifier (EDFA). **b** Quantum memory based on an $^{167}Er^{3+}$: $Y_2SiO_5$ crystal using the atomic-frequency combs (AFC) protocol, prepared with the polarization laser and the AFC laser. The AFC laser is locked to a Fabry–Pérot cavity (F–P cavity) using the Pound–Drever–Hall (PDH)

technique. The $^{167}Er^{3+}$: $Y_2SiO_5$ crystal is placed in a dilution refrigerator equipped with a superconducting magnet. The time sequence of photons and lasers is controlled by the optical switch, optical shutter and acousto-optic modulator (AOM), see text and Supplementary Note 1 for details. **c** Time-bin qubit entanglement analyzer consists of two unbalanced Franson interferometers and three Superconducting single-photon detectors (SSPD). The interferometers are phase stabilized with proportional-integral-derivative (PID) controllers by using a fraction of pump laser as reference light (Ref).

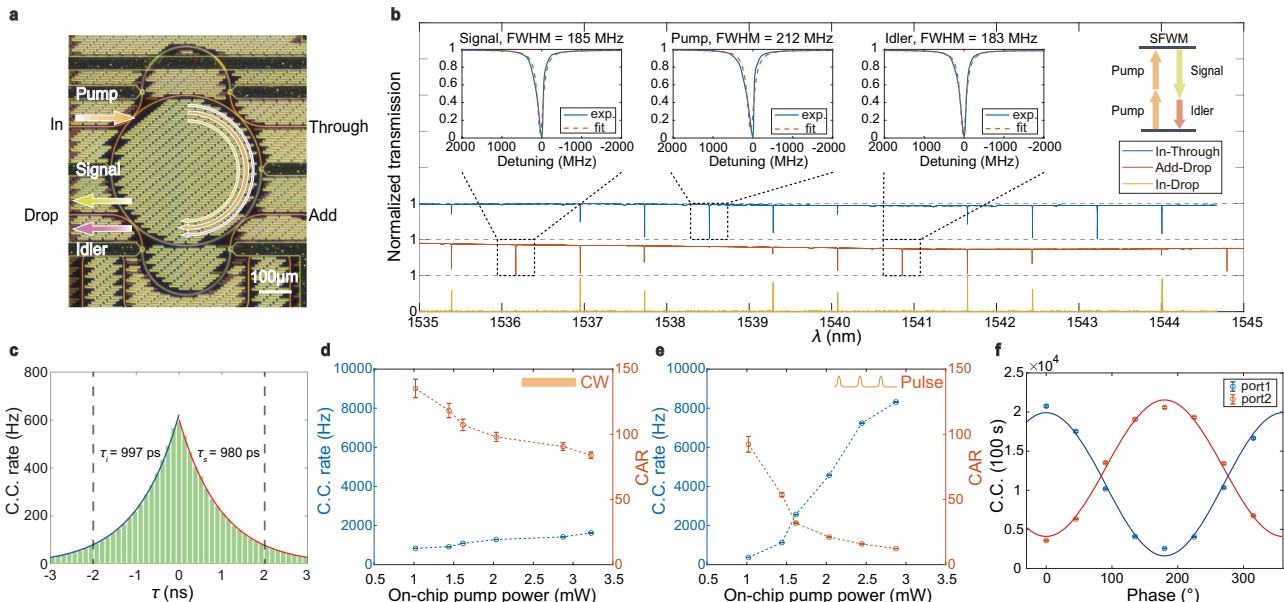

**Fig. 2 | Entangled photon-pair source based on integrated photonics. a** Optical micrograph of the integrated silicon nitride (SiN) dual Mach-Zehnder interferometer microring resonator (DMZI-R) for generating pairs of time-bin entangled photons. **b** Transmission spectra of the SiN DMZI-R source for In-Through, Add-Drop and In-Drop as input-output ports, shown in blue, red and yellow, respectively. Insets: The experimental spectra (blue) and the fitting Lorentzian functions (red) for the signal, pump and idler photons. Their full widths at half-maximum (FWHM) are about 185 MHz, 212 MHz, and 183 MHz, respectively. Upper-right: Two identical pump photons generate a pair of frequency non-degenerate signal and idler photons via the spontaneous four-wave mixing process. **c** Typical coincidence histogram with a 2.9-mW pulsed pump, shows a slight asymmetric feature,

resulting from different linewidths of signal and idler photons. The left (blue) and right (red) decay times are around 997 ps and 980 ps, respectively. We chose 4 ns as the coincidence window, indicated with two dashed vertical lines. **d, e** The coincidence counts (C.C.) rate (blue) and coincidence-to-accidental ratio (CAR, red) for continuous wave (CW)/pulsed pump (with 4-ns on-time and 32-ns period) as a function of average pump power. **f** Two-photon coincidence counts as functions of the phase between two Franson interferometers with a 2.9-mW pulsed pump. The blue/red circles are the raw data of coincidence counts between idler and port 1/2 of the signal photon, respectively. Their visibilities are 78.0 ± 0.4% and 70.5 ± 0.5%. See text for details. Error bars are derived from Poissonian statistics and error propagation.

Cavity-enhanced spontaneous parametric down-conversion has been employed as photon-pair source for rare-earth quantum memory, providing narrow-band photons compatible with the memory in the absence of any in-band narrow-band filters[70]. The transmission spectrum of this DMZI-R source is shown in Fig. 2b. The quality factors of the signal (about 1536 nm), pump (about 1538 nm) and idler (about 1540 nm) wavelengths are around $1.05 \times 10^6$, $0.92 \times 10^6$, and $1.06 \times 10^6$, respectively. Cavity-enhanced spontaneous four-wave mixing (SFWM) processes enable us to generate high-quality telecom entangled photons with high brightness and narrow linewidth, which are the essential requirements for telecom-compatible quantum memory. Figure 2c shows the coincidence histogram for a 2.9 mW pulsed pump, with a coincidence window of about 4 ns. The full width at half-maximum (FWHM) of the coincidence peak is 1370 ps, which corresponds to a coherence time of 997 ps and 980 ps for the idler and signal photons, respectively[71,72] (see Supplementary Note 3 for details). These photon-pair coincidence results are consistent with the linewidth of the classical transmission spectrum. To characterize our source, we further measure the coincidence counts (C.C.) rate and the coincidence-to-accidental ratio (CAR) as a function of average pump power with continuous wave (CW) and pulsed pump, shown in Fig. 2d, e, respectively. When we increase the pump power, due to the nature of SFWM, on one hand, we observe the enhancement of coincidence counts; on the other hand, high-order photon pairs reduce the CAR.

In the pulsed-pump case, the output of pump laser is chopped with the IM into pulses with a 4-ns pulse duration and a 32-ns period. By using these pump pulses, we generate time-bin entangled qubit state:

$$|\Phi\rangle = |ee\rangle + |ll\rangle, \tag{1}$$

in which $|e\rangle$ and $|l\rangle$ represent early and late temporal modes of the single photons, respectively. Photon pairs generated by the DMZI-R are time-bin entangled and the pair-creation time is uncertain within the coherence time of the pump laser. In order to analyze the two-photon entanglement, we send them into two Franson interferometers[73]. Each interferometer has two unbalanced optical paths, where the temporal difference between two optical paths is ~32 ns. Note that there are two time parameters that need to be treated carefully: the first one is the time difference between optical paths of one interferometer, $\Delta T$, which needs to match the period of the pump pulses and be larger than the single-photon coherence time; the second one is the difference between this parameter for two interferometers, $\Delta T_1 - \Delta T_2$, which needs to be smaller than the coherence time of the signal and idler photons. We fulfill both requirements by adjusting the pump-pulse period and accurate fiber splicing for both Franson interferometers. We scan the phase of the two unbalanced interferometers by tuning the setpoints of the two proportional-integral-derivative (PID) phase-locking systems for them, and obtain the two-photon quantum interference for the full $2\pi$ period, as shown in Fig. 2f. For entanglement analysis, we use the average pulsed-pump power of about 2.9 mW, where the CAR decreases to about 12.3. The visibilities ($V = \frac{\max - \min}{\max + \min}$) for these two curves are 78.0 ± 0.4% and 70.5 ± 0.5%. The nonideal visibilities are mainly due to the high-order photon pairs generated from the SFWM process, and the phase fluctuations of the Franson interferometers. To confirm the high-order photon-pair emission's impact on visibilities, we reduce the average power of the pulsed pump to 1.4 mW, and obtain visibilities of 84.8 ± 0.3% and 86.4 ± 0.3% (see Supplementary Note 2 for details). The phase fluctuations of the Franson interferometers are mainly due to the frequency instability of the partial pump laser used as the reference for PID locking (Fig. 1c).

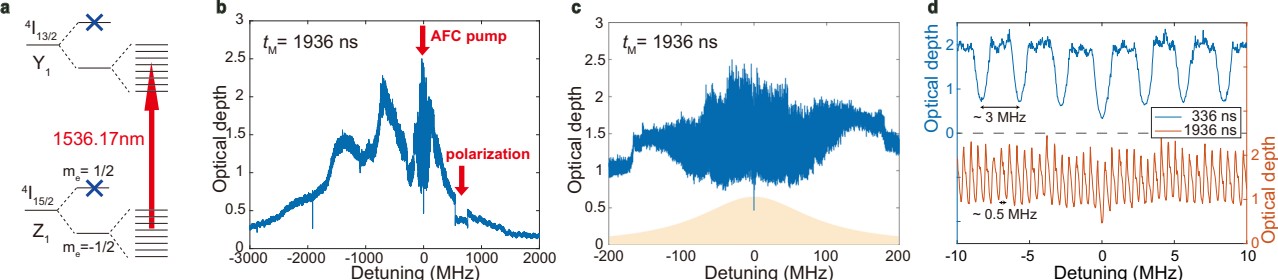

**Fig. 3 | An atomic-frequency-comb (AFC) quantum memory based on a $^{167}Er^{3+}$: $Y_2SiO_5$ crystal. a** The energy-level diagram of $^{167}Er^{3+}$: $Y_2SiO_5$. The Zeeman upper states are frozen under our experimental conditions. See text for details. **b** The full absorption spectrum after AFC preparation for 1936 ns storage time. Arrow AFC pump: central frequency of the AFC pump laser, denoted as 0 MHz for the detuning. Arrow polarization: the polarization laser scans from about 550–750 MHz detuning to enhance absorption. **c** The AFC spectrum for 1936 ns storage time matches the spectrum of the input signal photon (orange). **d** The zoom-in views of AFC spectrum for 336 ns (upper blue) and 1936 ns (lower red) storage times with frequency periods of about 3 MHz and 0.5 MHz, respectively.

## Quantum memory

The quantum memory is a 50 ppm doped $^{167}Er^{3+}$: $Y_2SiO_5$ crystal, which is cut along its $D_1$, $D_2$, and $b$ axes with dimensions of $4 \times 5 \times 9$ mm³. A photon-echo-type interaction using an atomic-frequency-comb (AFC)[74] enables the quantum state to transfer between the single photon and the ensemble of $^{167}Er^{3+}$ ions. To create an AFC, we use the optical pumping technique to shape the absorption profile of the ionic ensemble into a comb-like structure. The input photon is then absorbed and re-emitted into a well-defined spatial mode because of a collective rephasing of the ions in the AFC. The period of the comb determines the re-emission time and can be reconfigured.

Here we use the $^{167}Er^{3+}$ ions at site 1, with the $^4I_{15/2} \rightarrow {}^4I_{13/2}$ transition at ~1536.17 nm. $^{167}Er^{3+}$ exhibits eight hyperfine spin states due to a nuclear moment of $I = 7/2$ (see Fig. 3a). The crystal is placed inside a dilution refrigerator equipped with a superconducting magnet. The pump laser and signal photons propagate along the $b$ axis of the crystal. A magnetic field of about 1.5 T is applied in the $D_1$–$D_2$ plane with an angle of about $\theta = 120°$ to the $D_1$ axis for a large ground-state Zeeman splitting. The crystal's temperature is about 230 mK, deduced from the thermal-equilibrium population between Zeeman ground levels as a function of magnetic field strength (see Supplementary Note 4 for details). In this case, the ground-state Zeeman splitting is about 50 times $k_B T$, so that the electron spin is frozen to $m_e = -1/2$, ensuring long lifetimes and coherence times[51].

To improve the storage efficiency, a polarization laser is applied to enhance the absorption depth by polarizing the nuclear spin before AFC preparation. The polarization is performed partially because the transitions of $\Delta m_I = \pm 1$ and $\Delta m_I = 0$ cannot be resolved[55,57]. This is because the inhomogeneous broadening of our sample is larger than the hyperfine splitting. The ions are pumped away from "on" hyperfine states (resonant with the laser) and stored on "off" hyperfine states (off-resonant with the laser). Consequently, the population is still distributed over eight hyperfine states, but the spectrum profile of optical absorptions has been changed. We show the whole absorption spectrum in Fig. 3b, in which the polarization laser scans from about 550–750 MHz blue shifted from the center of the AFC, creating a pit there and consequently enhancing the absorption depth of the AFC. In our experiment, the magnetic field slightly deviates from the $D_1$–$D_2$ plane, lifting the degeneracy of the two classes of $^{167}Er^{3+}$ ions. To create the desired AFC profiles, we modulate the intensity of the AFC laser into pulses with a period equal to the storage time $t_M$. Its frequency spectrum will be a comb with $1/t_M$ period modulated by a sinc function whose main peak's envelop spans about 200 MHz. A close view of AFC is shown in Fig. 3c, compared with the spectral lineshape of the input signal photon, which shows the bandwidth matching between AFC and the signal photon. In Fig. 3d, we show the zoom-in views of AFC spectrum for 336 ns (upper blue) and 1936 ns (lower red) storage times with frequency periods of about 3 MHz and 0.5 MHz, respectively.

In the lower panel of Fig. 3d, we obtain a lower AFC contrast for data of 1936 ns. One limitation is the narrowest tooth width of AFC, which is about 0.25 MHz in this work. When the AFC's period is comparable with or narrower than the tooth width, the AFC contrast will be limited. The narrowest spectral features could be limited by residual frequency jitter of the AFC laser and power broadening in spectral hole burning processes. For the details on the optimization of storage time, see Supplementary Note 5.

In order to store the heralded single photon, we align the central frequency of the signal photon with respect to that of the quantum memory. To achieve that, we first coarsely tune the frequency of the photon-pair source by adjusting the temperature of the DMZI-R source. The DMZI-R resonance is sensitive to temperature, which is monitored with a thermistor, and stabilized with a thermoelectric cooler with feedback control. The resonance between the pump laser and the DMZI-R is maintained with a feedback loop by monitoring the remaining pump power at the through port[65]. Another knob we have is the magnetic field applied to the quantum memory. We vary the magnetic field amplitude to tune the Zeeman splitting and hence the optical transition frequency of $^{167}Er^{3+}$ ions. We present a typical result for the frequency alignment in Fig. 4a. We show the single counts (S.C.) of the signal (blue) and idler (red) photons as a function of the magnetic field amplitude. The clear drop of counts of signal photon shows the matching of the absorption frequency of the memory with respect to that of the signal photon. However, it is also important to have a stable count rate for idler photon, as the drop of signal-photon counts can also come from the frequency drift of the pump laser with respect to the DMZI-R. Only by observing the drop of signal-photon counts and, simultaneously, the constant idler-photon counts, we are certain that frequency alignment of the resonance of $^{167}Er^{3+}$ ions and the signal photon has been achieved. For a more precise alignment of signal photon to the AFC, we scan the pump laser's frequency for SFWM generation and the voltage on the on-chip resistors in finer steps to tune the frequency of the signal photon for achieving maximal storage efficiency, with the magnetic field fixed to 1.5 T. A typical result of 1936 ns storage time is shown in Fig. 4b.

After achieving the frequency alignment, we proceed with storage of the heralded signal photons. The time sequence of signal photons and lasers is controlled by the optical switch and optical shutter, with a 1.8-s polarization window, a 1.9-s AFC pump window, a 0.2 s delay and a 1-s memory window. An acousto-optic modulator (AOM) is inserted to gate the idler photons in synchronization with the memory window (see Supplementary Note 1 for details). In Fig. 4c, we show the coincidence histograms for different predetermined storage times of 1296 ns, 1616 ns, and 1936 ns. The 32-ns period of the side peaks is due to the periodic pulses of the pump laser, which creates accidental coincidence (AC) counts. The cross-correlation function is calculated as $g_{si}^2(0) = \frac{p_{si}}{p_s p_i}$, where $p_{si}$ is the probability of coincidence detections of

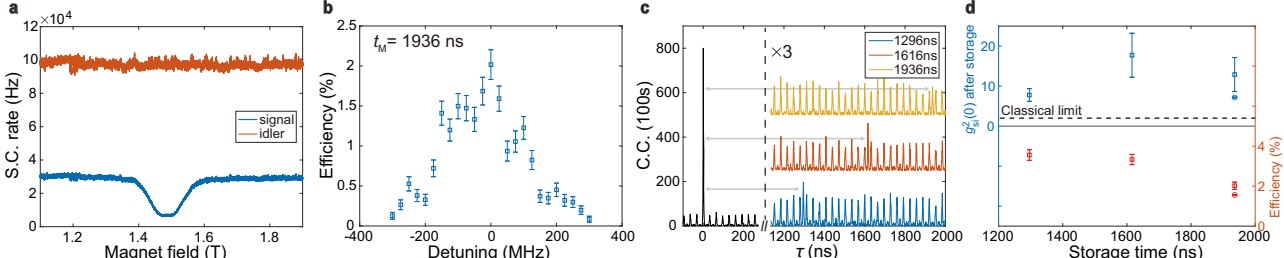

**Fig. 4 | Quantum storage of the heralded single photon in an $^{167}Er^{3+}$: $Y_2SiO_5$ crystal. a** Frequency alignment between the input signal photon and the absorption of $^{167}Er^{3+}$ ions. The single counts (S.C.) of the signal (blue) and idler (red) photons are shown as a function of the magnetic field amplitude. The clear drop of the counts of signal photon shows the matching of the absorption frequency of the memory with respect to that of the signal photon. **b** We scan the pump laser's frequency for SFWM generation in finer steps to tune the frequency of the signal photon for achieving maximal storage efficiency, with the magnetic field fixed to

1.5 T. **c** The coincidence counts (C.C.) histograms for storage times of 1296 ns (blue), 1616 ns (red) and 1936 ns (yellow), vertically offset for clarity. The data beyond 1200 ns are magnified by a factor of 3. **d** The second-order correlations (blue) and storage efficiencies (red) as functions of storage time, integrated in 100 s (square) and 40,000 s (circle). The second-order correlations stay well above the classical limit, $g_{si}^2(0) = 2$. Error bars are derived from Poissonian statistics and error propagation.

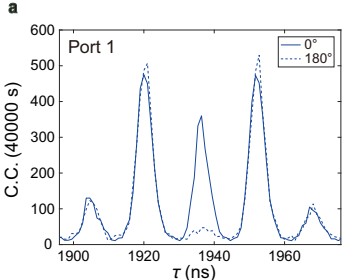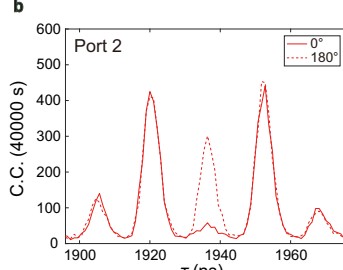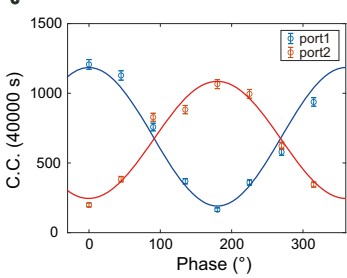

**Fig. 5 | Quantum storage of entanglement. a** After a storage time of 1936 ns, the signal photon is re-emitted from the quantum memory. By using two Franson interferometers for the re-emitted signal photon and the not-stored idler photon, we select the central peak (at 1936 s) to verify the entanglement between them. The coincidence counts (C.C.) between idler photons and port 1 of the Franson interferometer for signal photons are shown in solid/dashed blue curves for a phase of 0/180°, respectively. **b** The coincidence counts between idler photons and port 2 of the Franson interferometer for signal photons are shown in solid/dashed red curves

for the phase of 0/180°, respectively. Complementary counts at the same phase setting are clearly seen from in (**a**, **b**), as expected for interferometric measurements. **c** Full-period interference fringes show the coherence of the entangled state of Eq. (1). We scan the phase of the two Franson interferometers and obtain two-photon quantum interference for the full $2\pi$ period. The coincidence counts between port 1/2 of the Franson interferometer for signal photons and idler photons are shown in blue/red curves, respectively. Error bars are derived from Poissonian statistics and error propagation.

idler and signal photons, and $p_s$ ($p_i$) are the probabilities of single detections of signal (idler) photons, respectively. As shown in Fig. 4d, the cross-correlation remains well above the classical bound of thermal light[75,76], $g_{si}^2(0) = 2$. The memory efficiency is calculated as

$$\eta = \frac{echo}{input} = \frac{C.C.(echo)}{C.C.(transmission)} \times \frac{S.C.(transmission)}{S.C.(input)} \quad (2)$$

where C.C. (echo) and C.C. (transmission) are the respective coincidence counts at $\tau = t_M$ and $\tau = 0$. S.C. (transmission) and S.C. (input) are the respective single-photon counts with/without the AFC's absorption, neglecting the contribution of the retrieved photons. The memory efficiencies for various storage times and experimental settings are shown in Fig. 4d, indicating an efficiency of about 2% for 1936 ns storage. Note that all results presented in this article are without any subtraction of background noise or AC counts.

The maximum number of temporally multiplexed modes is approximately equal to the time-bandwidth product (TBP)[74], which is the storage time divided by the time width of the stored photons. In this work, the TBP is about 1936 ns/4 ns = 484. The realized mode number is 60.5, with 32 ns repetition time and 1936 ns storage time. One limitation here is the 32 ns repetition time, which is equal to the time difference of each unbalanced interferometer and should be larger than the single-photon coherence time, as discussed above. This can be solved by either coding the qubits on other degrees of freedom

(such as polarization), or still using time-bin qubits but with time multiplexing, in which the "early" mode of $N$ time-bin qubits can be inserted before the the "late" mode of the first qubit, thus enhancing the mode capability by a factor of $N$[77].

## Entanglement storage

After showing the storage of the heralded single photon, we now present the entanglement-storage results, in which the entanglement of the photon pair is preserved after the signal photon has been stored in the crystal for 1936 ns. To do so, we sent the stored and re-emitted signal photon and the not-stored idler photon into the entanglement analyzer (Fig. 1c), consisting of two unbalanced interferometers, for performing the Franson-type quantum interference experiment. In Fig. 5a, we show the histogram of coincidence between output port 1 for the signal photon and the output port for the idler photon. There are five peaks spanning from 1900 to 1980 ns. The central peak corresponds to the quantum interference peak of the entangled state of Eq. (1). By setting the phase of the two unbalanced interferometers in the entanglement analyzer to 0° and 180°, we observe fully constructive and destructive interference. The contrast of the central peaks are 76.0 ± 1.8%. In Fig. 5b, the coincidence between output port 2 for the signal photon and the output port for the idler photon is reversed for the same phase setting as in Fig. 5a. This is due to the complementary properties of the interferometers. The contrast of the central peaks are 68.5 ± 2.1%. The outmost two peaks in Fig. 5a, b are

**Table 1 | Entanglement witness**

| $t_M$(ns) | Signal port | $V$ | $g_{si}^2(0)$ | $\langle W \rangle$ |
|---|---|---|---|---|
| 1936 | 1 | 76.0 ± 1.8% | 7.16 ± 0.10 | −0.271 ± 0.009 |
|  | 2 | 68.5 ± 2.1% |  | −0.234 ± 0.010 |

Twofold coincidence visibility ($V$), the second-order correlation function ($g_{si}^2(0)$), and the expectation value of the entanglement witness ($\langle W \rangle$) for storage time $t_M$ = 1936 ns. The coincidence is between the idler photon and one of output ports of the signal photon, port 1/2, see Fig. 1c.

phase insensitive, as they correspond to the $|el\rangle$ and $|le\rangle$ coincidence counts, which are distinguishable in time. The other two peaks are the accidental coincidence peaks mentioned above. The results in Fig. 5a, b are integrated in 40,000 s. Furthermore, we scan the phase of the two unbalanced interferometers by tuning the setpoints of the two PID phase-locking systems (see Supplementary Note 7 for details), and obtain two-photon quantum interference for the full $2\pi$ period at 1936 ns storage time, as shown in Fig. 5c.

An entanglement witness[78] is employed to determine whether entanglement exists. The existence of entanglement is proved by determining a negative expectation value of the witness operator. The witness used here is given by

$$\hat{W} = \frac{1}{2} \left( |z^+z^-\rangle\langle z^+z^-| + |z^-z^+\rangle\langle z^-z^+| + |x^+x^-\rangle\langle x^+x^-| + |x^-x^+\rangle\langle x^-x^+| \\ - |y^+y^-\rangle\langle y^+y^-| - |y^-y^+\rangle\langle y^-y^+| \right), \quad (3)$$

where $|z^+\rangle = |e\rangle$, $|z^-\rangle = |l\rangle$, $|x^\pm\rangle = |e\rangle \pm |l\rangle$ and $|y^\pm\rangle = |e\rangle \pm i|l\rangle$[78,79]. Projections onto the $X-Y$ plane can be obtained from the visibility of the Franson interference. Combined with the results for Z basis (Fig. 4d) when the unbalanced interferometers are removed, the expectation value of the witness can be calculated using (see Supplementary Note 8 for detailed derivation):

$$\langle W \rangle = \frac{1}{g_{si}^2(0) + 2} - \frac{V}{2}. \quad (4)$$

The results of entanglement witness measurements are shown in Table 1, where all values are below the separable boundary of 0, by more than 23 standard deviations. Therefore, our results unambiguously show the successful storage of entangled photons in a crystal for 1936 ns.

## Discussion

Our results represent a significant advancement of quantum memory system. First, we show the storage of entangled photons in $^{167}Er^{3+}$: $Y_2SiO_5$, a quantum memory at telecom wavelength, which is a promising candidate for realizing an efficient, long storage time and broadband quantum memory[51,55–57]. In our work, we have extended the storage time of entangled photons at telecom wavelength more than 387 times longer than previous work[48]. Second, we show a successful combination of quantum memory with an integrated quantum-entanglement source, which generates narrow-band entangled photons, is CMOS compatible and hence suitable for scalable fabrication, and easier to use than sources based on bulk optics. We emphasize that the scalable entangled photon-pair sources we use here are essential to absorptive quantum memory. Unlike emissive quantum memories[80,81], each absorptive quantum memory requires an entangled photon-pair source. A scalable platform for entangled photon-pair sources is particularly important when a quantum network with multiple entangled photon sources and quantum memories with spatial mode multiplexing are constructed.

Despite these important results, there are several aspects that need to be further improved in our system. To improve the storage efficiency, one could optimize the AFC parameters[74,82] with more efficient initialization protocols with a resolved, long-lived hyperfine structure[51,56], and enhance the light-matter interactions with impedance-matched cavity[83] and integrated nano-structures[55,57,84–86]. The atomic-frequency comb spin wave protocol will facilitate the realization of a quantum memory with longer storage time and on-demand readout. One such possible protocol was recently demonstrated with Kramers ions in $^{171}Yb^{3+}$: $Y_2SiO_5$[53]. The level structure of $^{167}Er^{3+}$ is more complicated than $^{171}Yb^{3+}$, which requires more complicated optical pumping and coherent control pulse sequences. On the other hand, the rich level structure of $^{167}Er^{3+}$ may also bring new possibilities in quantum engineering of photon-atom interactions. With these experimental improvements, we anticipate $^{167}Er^{3+}$ ions and integrated quantum photonics to become a versatile platform for high-performance quantum memory, enabling the realization of large-scale quantum networks. We note that during the completion of this project, related work has shown the storage of heralded single photons in Er-doped fiber for up to 230 ns[87].

## Data availability

The data that support the plots within this paper and other findings of this study are available at https://github.com/NJU-Malab/Entanglement-Storage.

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

## Acknowledgements
The authors thank Kaixuan Zhu, Penghong Yu, and Xuntao Wu for experimental helps during the early stage of this work. This research was supported by the National Key Research and Development Program of China (Grants Nos. 2022YFE0137000, 2019YFA0308704, and 2017YFA0303704), the National Natural Science Foundation of China (Grants Nos. 11690032 and 11321063), the NSFC-BRICS (Grant No. 61961146001), the Leading-Edge Technology Program of Jiangsu Natural Science Foundation (Grant No. BK20192001), the Fundamental Research Funds for the Central Universities, and the Innovation Program for Quantum Science and Technology (Grants Nos. 2021ZD0300700 and 2021ZD0301500).

## Author contributions
M.-H.J., W.X., and X.-S.M. designed and performed the experiment. Q.H., Y.-Y.A., X.Z., W.-J.X., and Y.-B.X. provided experimental assistance and suggestions. M.-H.J., W.X., and X.-S.M. analyzed the data. M.-H.J., W.X., Q.H., and X.-S.M. wrote the manuscript with input from all authors. Y.L., S.Z., and X.-S.M. supervised the project.

## Competing interests
The authors declare no competing interests.
