## [Peer Review File · Nature Communications]

Quantum storage of entangled photons at telecom wavelengths in a crystalEditorial Note: This manuscript has been previously reviewed at another journal that is not operating a transparent peer review scheme. This document only contains reviewer comments and rebuttal letters for versions considered at *Nature Communications*.

REVIEWERS' COMMENTS

Reviewer #1 (Remarks to the Author):

I believe authors have completely addressed my concerns and questions. In light of the new results, I strongly recommend publication of the current manuscript in Nature Communication. Given the difficulties of working Erbium ions and importance of telecom photon storage, this is a significant result and it is an important demonstration of source-memory integration

Reviewer #3 (Remarks to the Author):

After the authors clarified several points regarding the significance of their work, I believe their intention is to claim that this is the first demonstration of entangled photons in the 167ErYSO system, a promising material for quantum networks. I appreciate the importance of introducing entangled photons as a source, and further improving memory efficiency. I think that the manuscript is appropriate for Nature Communications. However, I have some suggestions that should be addressed before the manuscript can be published in Nature Communications.

To reach a broader audience and enhance the manuscript's readability, I recommend the authors rephrase the paper in a less technical manner. There are several points, among others, that make this paper sound overly technical and specific. For instance, the entire introduction mainly focuses on the work within the rare-earth ions community, explains non-Kramers ions and kramers ions, and eventually lands on 167Er . This level of detail may be too specific for a general audience. Additionally, the beginning of the experiment section immediately delves into the schematics of the setup, whereas much of this content could be

incorporated into the caption of the first figure.

I have two additional technical comments:

1. In Figure 2e, the authors examined the dependence of coincidence counts rate (C.C.) and coincidence-to-accidental ratio (CAR) on the average pump power. They selected 2.9 mW to achieve the maximum C.C. and minimum CAR. However, it appears that the curves did not saturate. I wonder why the authors did not consider increasing the pump power further to explore this aspect.

2. In Figure 3d, the contrast of the teeth is approximately 50-60%. This suggests the presence of some ions acting as absorbing ions, which ideally should be minimized. Additionally, the lower panel exhibits lower contrast compared to the upper panel. It would be helpful if the authors could comment on the limitations in their experiments. Typically, this depends on the ratio of spin lifetime to optical lifetimes.

We would like to thank the reviewers for their valuable comments and suggestions. We addressed all the issues raised by the reviewers, which helped to improve the readability of our manuscript. Please find below details of the specific changes made, in a point-by-point response. Reviewer's remarks are in *blue italic font*, our replies in black normal font, and new text in **red normal font**. In the revised manuscript, we have added line numbers and highlighted new text in **red font**.

Reviewer #1 (Remarks to the Author):

I believe authors have completely addressed my concerns and questions. In light of the new results, I strongly recommend publication of the current manuscript in Nature Communication. Given the difficulties of working Erbium ions and importance of telecom photon storage, this is a significant result and it is an important demonstration of source-memory integration.

We thank the reviewer for his/her positive assessments of our work.

Reviewer #3 (Remarks to the Author):

After the authors clarified several points regarding the significance of their work, I believe their intention is to claim that this is the first demonstration of entangled photons in the $^{167}\text{ErYSO}$ system, a promising material for quantum networks. I appreciate the importance of introducing entangled photons as a source, and further improving memory efficiency. I think that the manuscript is appropriate for Nature Communications. However, I have some suggestions that should be addressed before the manuscript can be published in Nature Communications.

We thank the reviewer for his/her positive assessments of our work.

To reach a broader audience and enhance the manuscript's readability, I recommend the authors rephrase the paper in a less technical manner. There are several points, among others, that make this paper sound overly technical and specific. For instance, the entire introduction mainly focuses on the work within the rare-earth ions community, explains non-Kramers ions and Kramers ions, and eventually lands on ^{167}Er . This level of detail may be too specific for a general audience. Additionally, the beginning of the experiment section immediately delves into the schematics of the setup, whereas much of this content could be incorporated into the caption of the first figure.

We thank the reviewer for these suggestions and revise our manuscript accordingly. In the Introduction section, we have now greatly simplified the

details of non-Kramers ions and Kramers ions. In the Experimental setup section, we give a concise summary of the experiment before detailed schematics and incorporate technical contents of our experiment schematics into the caption of the first figure. The changes are highlighted in red font in the revised manuscript.

I have two additional technical comments:

1. In Figure 2e, the authors examined the dependence of coincidence counts rate (C.C.) and coincidence-to-accidental ratio (CAR) on the average pump power. They selected 2.9 mW to achieve the maximum C.C. and minimum CAR. However, it appears that the curves did not saturate. I wonder why the authors did not consider increasing the pump power further to explore this aspect.

Reply 3.1: We thank the reviewer for raising this point. There is a trade-off between C.C. and CAR: a higher pump power will result in a lower CAR and limit the visibility of Franson interference, as supported by Supplementary Note 2. Technically, high pump power will shift the micro-ring's resonance by thermo-optic effect and hence reduce the stability of the entangled photon source. Therefore, in our work, we didn't use more than 2.9 mW pump power.

In line 137, we write: “The non-ideal visibilities are mainly due to the high-order photon pairs generated from SFWM process, and the phase fluctuations of the Franson interferometers. To confirm the high-order photon-pair emission's impact on visibilities, we reduce the average power of the pulsed pump to 1.4 mW, and obtain visibilities of $84.8 \pm 0.3\%$ and $86.4 \pm 0.3\%$ (see Supplementary Note 2 for details).”

2. In Figure 3d, the contrast of the teeth is approximately 50-60%. This suggests the presence of some ions acting as absorbing ions, which ideally should be minimized. Additionally, the lower panel exhibits lower contrast compared to the upper panel. It would be helpful if the authors could comment on the limitations in their experiments. Typically, this depends on the ratio of spin lifetime to optical lifetimes.

Reply 3.2: We thank the reviewer to point this out. The AFC contrast in this experiment is limited by the bandwidth of our AFC. For a narrower bandwidth (below 20 MHz), we have observed higher AFC contrast (~80%). For a wider AFC (~200 MHz in our work), the AFC laser's power is divided to different and more frequencies. Therefore, the power of each individual frequency tooth and hence the AFC contrast are both reduced. This limiting factor can be solved with the integrated nano-structures which concentrate the laser power and enhance the light-matter interactions.

In line 288, we write: “To improve the storage efficiency, one could optimize the

AFC parameters [75,83] with more efficient initialization protocols with a resolved, long-lived hyperfine structure [51,57] and enhance the light-matter interactions with impedance-matched cavity [84] and integrated nano-structures [56,58,85-87].”

For the lower panel of Fig. 3d, an additional limitation is the narrowest tooth width of AFC, which is about 0.25 MHz in this work. When the AFC's period is comparable with or narrower than the tooth width, the AFC contrast will be limited. The narrowest spectral features could be limited by residual frequency jitter of the AFC laser and power broadening in spectral hole burning process.

In line 181, we write: “In the lower panel of Fig. 3d, we obtain a lower AFC contrast for data of 1936 ns. One limitation is the narrowest tooth width of AFC, which is about 0.25 MHz in this work. When the AFC's period is comparable with or narrower than the tooth width, the AFC contrast will be limited. The narrowest spectral features could be limited by residual frequency jitter of the AFC laser and power broadening in spectral hole burning processes.”